# Enhanced Adsorptivity of Hexavalent Chromium in Aqueous Solutions Using CTS@nZVI Modified Wheat Straw-Derived Porous Carbon

**DOI:** 10.3390/nano14110973

**Published:** 2024-06-03

**Authors:** Tiantian Deng, Hansheng Li, Su Ding, Feng Chen, Jingbao Fu, Junwei Zhao

**Affiliations:** 1School of Environmental and Biological Engineering, Henan University of Engineering, Zhengzhou 451191, China; hansonlee1989@gmail.com (H.L.); sue_ding189@163.com (S.D.); chenfeng871588@163.com (F.C.); fujingbao@126.com (J.F.); 2Faculty of Health Sciences, University of Technology MARA, Puncak Alam Campus, Puncak Alam 42300, Malaysia; 3College of Resources and Environment, Yangtze University, Wuhan 434023, China; zjw882@139.com

**Keywords:** chromium, wheat straw biochar, nZVI, chitosan, adsorption, chemical reduction

## Abstract

Using KOH-modified wheat straw as the precursor, wheat straw biochar was produced through carbonization at 500 °C. Subsequently, a synthetic material containing nano-zero-valent iron (nZVI) was prepared via liquid phase reduction (nZVI-WSPC). To enhance its properties, chitosan (CTS) was used by crosslinking to form the new adsorbent named CTS@nZVI-WSPC. The impact of CTS on parameters such as mass ratio, initial pH value, and adsorbent dosage on the adsorption efficiency of Cr(VI) in solution was investigated through one-factor experiments. Isotherm adsorption and thermodynamic analysis demonstrated that the adsorption of Cr(VI) by CTS@nZVI-WSPC conforms to the Langmuir model, with a maximum adsorption capacity of 147.93 mg/g, and the adsorption process is endothermic. Kinetic analysis revealed that the adsorption process follows a pseudo-second-order kinetic model. The adsorption mechanism, as elucidated by SEM, FTIR, XPS, and XRD, suggests that the process may involve multiple mechanisms, including pore adsorption, electrostatic adsorption, chemical reduction, and surface chelation. The adsorption capacity of Cr(VI) by CTS@nZVI-WSPC remains high after five cycles. The adsorbent is simple to operate, economical, efficient, and reusable, making it a promising candidate for the treatment of Cr(VI) in water.

## 1. Introduction

The issue of heavy metal pollution in aquatic environments has intensified, posing a significant challenge due to the indiscriminate discharge of industrial waste, mining effluents, sewage irrigation, and chemical products [1]. Among these contaminants, chromium pollution stands out as a particularly pressing concern due to its inherent toxicity and deleterious effects on aquatic ecosystems [2]. Hexavalent chromium (Cr(VI)), in particular, poses a direct threat to human health. Exposure through ingestion, inhalation, or skin contact can result in severe toxicity, potentially causing genetic defects, allergies, and long-term damage to the environment [3].

Recent investigations have revealed that biomass materials, when subjected to high-temperature pyrolysis in an oxygen-deficient environment, transform into carbon-rich materials known as biochar (BC). Biochar is characterized by a wealth of functional groups and a large specific surface area, making it highly desirable for applications in aquatic environments due to its exceptional adsorption capacity. In the context of remediating chromium-containing wastewater, various types of biochar have demonstrated superior adsorption and removal capabilities. Common agricultural waste biomasses, including oat waste [4] sawdust of beech [5], rice husk [6], bagasse [7], wheat bran [8], coconut Husk [9], bamboo [10], and rubberwood fiber [11], have all demonstrated exceptional potential for application in both chromium pollution removal and degradation.

Proper modification of biochar can significantly enhance its adsorption performance in multiple ways [12]. A range of modification techniques have been investigated to optimize the adsorption capacity of hexavalent chromium [13]. In chemical oxidation or reduction modifications, the introduction of oxygen-containing functional groups such as -COOH and -OH onto the biochar surface serves to enhance its adsorption capacity. Commonly employed agents for such modifications include HCl and HNO_3_, as well as alkaline substances like KOH and NaOH [14]. Metal impregnation, characterized by the adsorption of metal ions or heteroatoms into the pores and surfaces of biochar, represents another effective modification strategy. This process increases the specific surface area of biochar, and the synergy between metal ions and adsorbents further enhances the adsorption capacity [15].

nZVI has garnered considerable attention in the scientific community due to its plethora of active sites, exceptional reactivity, and remarkable reduction efficacy in the reduction of Cr(VI) to Cr(III) [16]. However, its intrinsic magnetism, heightened reactivity, susceptibility to agglomeration, and susceptibility to oxidation pose challenges that markedly impact its performance. To surmount these challenges, there is a growing interest in investigating the amalgamation of nZVI with biochar [17,18]. The binding interactions between nanoscale zero-valent iron particles and biochar may encompass adsorption, coordination, and chelation mechanisms [19]. For instance, ferrous or iron ions can form bicrenate complexes by binding to -OH groups. Furthermore, the oxide shell encapsulating the nZVI core interacts with the surface functional groups of biochar, fostering the formation of robust and stable bonds. This interfacial interaction significantly enhances the stability and functionality of the composite material, facilitating its application in various environmental and energy-related fields [20]. These interactions diminish the strength of attraction between particles, consequently mitigating nZVI aggregation and promoting its dispersion within the biochar matrix.

In the adsorption reaction of hexavalent chromium, NH_2_ has demonstrated functional reactivity [21]. Chitosan, derived from the natural polysaccharide chitin, is a high-molecular-weight product known for its non-toxic and biocompatible properties. Chitosan is rich in amino (-NH_2_) and hydroxyl (-OH) functional groups, serving as robust binding sites for Cr (VI) and facilitating the adsorption of various pollutants through mechanisms such as adsorption, chelation, electrostatic attraction, and ion exchange [22,23]. When coupled with Fe-based materials, chitosan effectively addresses the inherent magnetic concerns, thereby enhancing the stability and adsorption capacity of the reaction. Notably, the synergistic combination of chitosan and biochar leads to the formation of novel carbohydrate polymers with significantly augmented adsorption capacity.

Building on the aforementioned synthesis, this study focused on the fabrication of a composite wheat straw biochar adsorbent. Chitosan was introduced for cross-linking with nanoscale zero-valent iron (nZVI) to evaluate its efficacy in adsorbing hexavalent chromium from aqueous solutions. The physical and chemical properties, along with the surface structure characteristics of the materials, were comprehensively analyzed using SEM, FTIR, XRD, BET, and XPS. Additionally, the effects of pH and adsorbent dosage on the adsorption of hexavalent chromium in water were systematically investigated to identify the critical factors influencing Cr(VI) removal. The study also delved into the dynamics of adsorption, isotherm analysis, and thermodynamic modeling to elucidate the mechanisms governing the adsorption behavior. The aim was to provide insights that can inform the development of functional materials for the treatment of chromium-containing wastewater.

## 2. Materials and Methods

### 2.1. Preparation of Materials

Wheat straw-derived porous: Wheat straws were collected from a cultivated field in Nanyang city, Henan Province, China. To eliminate impurities from the wheat straw, the materials were sieved through a 20-mesh screen and subsequently washed with distilled water before being dried in an oven at 80 °C for 24 h. An amount of 5 g of wheat straw powder was placed in 20 mL of KOH solution with a concentration of 2 mol/L and soaked for 12 h. Following the oscillation, the wheat straw powder underwent filtration and was rinsed with pure water until the pH of the filtrate reached neutrality. The resultant powder was dried at 105 °C for 3 h and stored for subsequent use. During the carbonization process, an appropriate quantity of the pre-obtained wheat straw powder was placed in a quartz boat and transferred to the quartz tube within a tube furnace. The gas tightness was ensured by nitrogen testing for 10 min, after which the heating program commenced: with a heating rate of 10 °C/min, the carbonization temperature was set to 500 °C, and the carbonization duration was 180 min. Upon completion of the carbonization process and the subsequent cooling to room temperature, the residue was retrieved and placed in a sealed beaker for preservation, denoted as the original biochar, referred to as WSPC.

nZVI-WSPC [24,25]: 2 g of wheat straw biochar (WSPC) was dispersed in 50 mL of FeSO_4_·7H_2_O solution with a concentration of 0.1 mol/L and stirred for 2 h. After that, 50 mL of aqueous ethanol solution with a volume ratio of 1:1 was added as a dispersant to a three-port reaction glass flask, followed by 50 mL of NaBH_4_ solution at a concentration of 0.25 mol/L at a rate of 2 drops/s under nitrogen. Approximately 30 min later, biochar-loaded nanoscale zero-valent iron particles were successfully synthesized. The resulting carbonized products underwent a triple wash with deionized water and anhydrous ethanol, followed by vacuum drying at 60 °C for 24 h. Subsequently, the products were ground through a 0.15 mm sieve and designated as nZVI-WSPC.

CTS@nZVI-WSPC: Mix 5 g of chitosan with 100 mL of acetic acid solution with a volume fraction of 5% at 25 °C and stir vigorously until the chitosan gel dissolves. Then, nZVI-WSPC was added with a mass ratio of chitosan: nZVI-WSPC (1:1), and continuous mechanical stirring for 4 h was used to obtain a uniform CTS@nZVI-WSPC mixture. Then, the mixed gel solution is sonicated in an ultrasonic bath for 30 min to obtain a homogeneous and dispersed chitosan gel solution with nZVI-WSPC particles. The prepared solution is lyophilized and dried for 28 h to obtain complexes. The final product is crushed and passed through a 100 mesh size (150 mm) for further use.

Those three adsorbents mentioned above were obtained as shown in Figure 1.

### 2.2. Adsorption Experiment

K_2_Cr_2_O_7_, precisely measured at 2.835 g, was dissolved in 1000 mL of deionized water to create a stock solution with a Cr (VI) concentration of 1 g/L. This stock solution was further diluted with deionized water as necessary. The pH of the reaction solution was adjusted using varying concentrations of HCl and NaOH solutions.

#### 2.2.1. Choice of Adsorbent

In order to find the optimal mass ratio of CTS and nZVI-WSPC, we set six different gradients, namely CTS: nZVI-WSPC = 0:1, 1:1, 1:2, 1:3, 3:1, 2:1. The experimental conditions at this time were C_0_ = 50 mg/L, T = 25 °C, t = 24 h, Dose = 1 g/L, pH = 7.0. Following single-factor comparative experiments, we identified the synthesized materials resulting from a mass ratio of CTS: nZVI-WSPC = 1:3 for subsequent characterization and batch experiments.

#### 2.2.2. Effects of Adsorbent Dose and Solution pH

Similarly, the adsorbent dose and pH value were also examined as important factors affecting the experimental results. In agreement with the experimental conditions described above, we studied the removal of Cr (VI) with different adsorbent doses and pH values. The amount of adsorbent was set to 0.2 g/L, 1 g/L, 2 g/L, 4 g/L, 8 g/L, and 10 g/L, while pH was set to 2, 3, 4, 5, 6, 7, 8, 9, 10, respectively.

#### 2.2.3. Adsorption Isotherm Experiment

The analysis of adsorption isotherms aims to ascertain the adsorption capacity of the three adsorbents for Cr (VI). The initial concentration range of Cr (VI) was set from 5 to 200 mg/L. The mass of the adsorbent added to the 50 mL Cr (VI) solution was 0.05 g, the reaction time was 24 h, the solution maintained a pH of 2, and the temperature of the reaction system was set at 25 °C, 35 °C, and 45 °C. The data were fitted using three widely recognized isothermal adsorption models, and the experimental results of the adsorption isotherm were thoroughly analyzed.

#### 2.2.4. Adsorption Kinetics Experiment

The adsorption kinetic experiments, along with the corresponding fitting equations, aim to elucidate the speed of the adsorption reaction and its underlying mechanism. Three distinct adsorbent materials underwent adsorption kinetics testing under identical reaction conditions. Specifically, 0.05 g of material was introduced to 50 mL of Cr (VI) solution with an initial concentration of 50 mg/L, while maintaining a reaction system pH of 2 and a temperature of 25 °C. Samples of the supernatant were extracted at specific intervals, including 5 min, 10 min, 20 min, 30 min, 50 min, 70 min, 90 min, 180 min, 240 min, 480 min, 960 min, and 1440 min. These samples were subsequently filtered to determine the concentration of Cr (VI).

#### 2.2.5. The Reusability of CTS@nZVI-WSPC

To evaluate the reavailability of CTS@nZVI-WSPC, we carried out repeatability tests under fixed experimental conditions. The experimental conditions were set to pH = 2.0, Cr (VI) = 50.0 mg/L, and dose = 1.0 g/L. After each reaction process, we recovered the solid-phase adsorbent CTS@nZVI-WSPC composite using a magnetic material, followed by washing and drying with purified water. The experiments were repeated five times.

## 3. Results and Discussion

### 3.1. Characterization

The scanning electron microscope was employed to examine the morphological characteristics of WSPC, nZVI-WSPC, and CTS@nZVI-WSPC materials. As depicted in Figure 2a, wheat straw biochar exhibits a distinctive, regular, and porous structure, with an average pore size distribution below 5 μm. The composite biochar displays notable morphological features. On the nZVI-WSPC surface Figure 2b, a uniform distribution of heterogeneous spherical particles is evident, indicating a dispersion of nZVI nanoparticles rather than aggregation on the WSPC. This dispersion significantly enhances the adsorption sites of nZVI-WSPC [26]. Figure 2c illustrates the morphological structure of CTS@nZVI-WSPC, revealing that the introduction of CTS renders the material surface rougher. This alteration enhances the dispersion and structural stability of n-ZVI on the WSPC surface. EDS results further confirm the composite’s composition, revealing the presence of C, O, and Fe. These findings affirm the successful loading of nano-zero-valent iron or iron oxide onto the biochar surface.

Raman spectroscopy is an effective means to demonstrate the structure of carbon materials. Both the D and G peaks are recognized as Raman characteristic features of carbon atomic crystals, located at around 1350 cm^−1^ and 1580 cm^−1^, respectively. The D peak reflects the carbon lattice defects, while the G peak signifies the material’s degree of carbonization [27]. In this study, I(D)/I(G) represents the area ratio of the D peak to the G peak, with a larger ratio indicating a higher level of defects in the carbon atomic crystal.

The characteristic peaks of the three materials were fitted using the Gauss model, yielding termination coefficients (R^2^) of 0.961, 0.972, and 0.963 for WSPC, nZVI-WSPC, and CTS@nZVI-WSPC, respectively. These values signify a robust degree of carbonization and prominent characteristic peaks in all three materials. As depicted in Figure 3a, the I(D)/I(G) ratios for WSPC, nZVI-WSPC, and CTS@nZVI-WSPC were 0.295, 0.415, and 0.434, respectively. This suggested an increasing degree of defects in the latter two composites, which is advantageous for providing more active sites for subsequent reactions.

The qualitative analysis of surface functional group species in three distinct materials using FTIR is presented in Figure 3b. All materials exhibit absorption peaks at 1095 cm^−1^ and 1632 cm^−1^, which are characteristic of the functional groups inherent to biochar itself [28]. For peak around 1632 cm^−1^ is linked to the C=O stretching vibration of ester and the tensile vibration of O-H, and the wavenumber at 1095 cm^−1^ signifies the vibration absorption peak of C-O [29,30]. Nanozero-valent iron minimally influences the surface functional groups of the biochar. Upon the introduction of CTS, a contraction absorption peak emerges within the wide peak range from 3328 to 3526 cm^−1^, indicating the presence of intramolecular and intermolecular hydrogen bonds in the CTS intramolecular hydroxyl and amino groups. The loading of CTS-nZVI preserved the structural stability of the biochar itself, allowing the C-H and C=C functional groups on the surface to directly or indirectly participate in and enhance the subsequent adsorption process.

Through the XRD characterization of the three materials, a distinctive diffraction peak appears near 20°, as illustrated in Figure 3c. Comparison with the PDF card, the characteristic peak suggested the presence of certain carbon material composition in the three materials [31]. The introduction of nZVI and CTS did not alter the structure and composition of the original biological carbon. No significant deviation was observed in the characteristic peaks of nZV-WSPC and CTS@nZVI-WSPC, indicating a minimal difference in their crystal structure. This is attributed to the fact that CTS acts as a natural polymer polysaccharide, with characteristic diffraction peaks predominantly occurring at 10–20°. The appearance of this characteristic peak is a result of intermolecular or intramolecular hydrogen bonds stemming from the presence of -OH and -NH_2_ functional groups in CTS [32,33]. Many salts or oxides tend to disperse on the carrier surface to form monolayers and submonolayers. In the case where the loading amount is below a certain threshold, a monolayer dispersion state is maintained, and the active component cannot be detected by X-ray. From the XRD profile, it is evident that nZVI was successfully synthesized in both composites. The diffraction peaks are located at 30.27°, 35.68°, 44.5°, 57.4°, and 63.25°, corresponding to different forms of iron (Fe), with the peaks at 44.5° and 63.25° specifically indicating the presence of nano-zero-valent iron, as reported in various literature sources [34]. Moreover, the diffraction peaks of CTS@nZVI-WSPC at 2θ = 29.9°, 40.17°, and 55.08° align well with the ferric oxide standard card, which appears after the adsorption reaction [35,36]. The characteristic peak band at 35.19° represents the formation of ferric oxide, suggesting that the composite may undergo oxidation after the reduction by iron particles, which is consistent with the XPS analysis results. The comprehensive analysis showed that the introduction of nZVI and chitosan did not alter the carbon structure of the raw wheat straw. After the reaction, the material’s structure remained largely unchanged, retaining a stable morphology.

As depicted in Figure 3d, the nitrogen adsorption-desorption isotherm of CTS@nZVI-WSPC exhibits a closed loop, which corresponds to I-type isotherms (IUPAC classification) and H4-type hysteresis loop [37]. According to the extended isotherm, the curve shows a steep rise followed by a plateau, indicative of micro-porous structure and monolayer reversible adsorption consistent with the results of the isothermal adsorption fitting. This adsorption generally occurs in microporous materials and mesoporous materials with pore sizes very close to micropores [18], which showed a strong agreement with the biochar pore structure observed by SEM.

### 3.2. Batch Experiments

#### 3.2.1. Effect of Ratio

The standard curve for hexavalent chromium determination was shown in Figure 4a. A series of adsorbents synthesized from chitosan (CTS) and nZVI-WSPC with varying mass ratios were employed for the comparative adsorption of hexavalent chromium in aqueous solutions, and the results are presented in Figure 4b. As depicted in the figure, at a constant adsorbent mass (dose = 0.05 g), nZVI-WSPC exhibits a more pronounced impact on the adsorption of Cr (VI) in water compared to CTS alone. The composite materials demonstrated a higher adsorption capacity for Cr (VI) in water than nZVI-WSPC alone, reaching a maximum value of 39.78 mg/g at the mass ratio m(CTS):m(nZVI-WSPC) = 1:3. However, with a continuous increase in the CTS content, the adsorption capacity of the material gradually decreased. The modification of nZVI-WSPC by chitosan proved beneficial for the adsorption of Cr (VI) in water. Excessive CTS may impede the contact of nano-ZVI or biochar with hexavalent chromium, thereby reducing adsorption efficiency [38]. Based on the obtained experimental data, the optimal mass ratio of chitosan to nZVI-WSPC was determined to be 1:3, and this ratio was employed in subsequent experiments.

#### 3.2.2. Effect of the Dose of Adsorbent

The quantity of adsorbent directly influences the adsorption effectiveness. Figure 4c illustrates the experimental results demonstrating the impact of varying material dosages on the adsorption capacity of Cr (VI) solution. Under the same reaction conditions, WSPC, nZVI-WSPC, and CTS@nZVI-WSPC all exhibited similar adsorption trends. As the dosage increased, the adsorption amount of Cr(VI) gradually decreased, while the adsorption rate steadily increased. This can be explained by the fact that when the solution concentration is constant, an increase in dosage leads to a proportional increase in the number of active adsorption sites available on the material [39]. However, as the adsorption of solutes in solution approaches saturation, an excess of active adsorption sites occurs, reducing the adsorption efficiency per unit mass of the material. Both modified materials showed improved adsorption properties to some extent. It is worth noting that the pH of the reaction system was not adjusted during these experiments, so the three adsorbents did not achieve their optimal performance.

#### 3.2.3. Effect of Initial pH

Under the same experimental conditions, the adsorption efficiency of Cr (VI) in water was compared under different initial pH conditions. The results are presented in Figure 4d. Across solutions with pH values ranging from 2 to 10, the adsorption of Cr (VI) by all three materials exhibited a decrease with increasing pH. At pH 2, the adsorption reached its maximum, with the adsorption capacities of WSPC, nZVI-WSPC, and CTS@nZVI-WSPC reaching 21.25 mg/g, 33.26 mg/g, and 43.28 mg/g, respectively—significantly higher than under neutral and alkaline conditions. This suggests that pH is a crucial factor influencing the removal properties of the materials.

Numerous studies have shown that Cr (VI) may have different ionic forms in aqueous solutions. In the acidic environment of pH < 5.0, the primary form is hydrogen chromate (HCrO_4_^−^), and with increasing pH, Cr (VI) mainly exists as chromate ions (CrO_4_^2−^). Since the adsorption-free energy of HCrO_4_^−^ is lower than that of CrO_4_^2−^, HCrO_4_^−^ at low pH is more adsorbed to CrO_4_^2−^ than at high pH [40]. In this study, WSPC underwent pretreatment modification using KOH; hence, the introduction of OH^−^ would lead to competitive adsorption under alkaline conditions, reducing the material’s adsorption capacity [41]. 

Under acidic conditions, the oxide layer on the surface of nZVI was disrupted, exposing more Fe^0^ sites for the reduction reaction of Cr (VI). However, under alkaline conditions, Fe tends to form a co-precipitate on the surface of nZVI-WSPC, leading to the passivation of the material surface and limiting the reaction. Furthermore, Cr (VI) has a high positive redox potential under acidic conditions (pH = 1, E0 = 1.3 V; pH = 5, E0 = 0.68 V), so the reaction between nZVI and Cr (VI) is very rapid [42]. In addition, the contact between nZVI and Cr (VI) also contributes to the iron corrosion caused by H^+^ under acidic conditions. Under neutral or alkaline conditions, Cr^3+^ and Fe^3+^ will form a precipitate covering the nZVI surface to rapidly inactivate it [43].

In contrast, CTS@nZVI-WSPC was less influenced by pH than the other two materials. On the one hand, the introduction of CTS enhances the mechanical stability of the material; on the other hand, under acidic conditions, the amino group (NH_2_) of chitosan is readily protonated to form (NH_3_^+^), acquiring a positive charge and facilitating adsorption with HCrO_4_^−^, resulting in increased anion adsorption by the adsorbent in acidic environments [44]. Under alkaline conditions, the decreased adsorption can be attributed to electrostatic repulsion on the deprotonated CTS@nZVI-WSPC surface and increased competition for more hydroxide ions. At the same time, it was obvious that CTS@nZVI-WSPC is less sensitive to pH than the other two materials, thus it can be suitable for the removal of Cr in a wide range of pH.

### 3.3. Adsorption Isotherm

In this study, CTS@nZVI-WSPC was employed to investigate the adsorption behavior and mechanism of Cr (VI). As depicted in Table 1 and Appendix A, the Langmuir model proved to be more effective in elucidating the adsorption behavior of Cr (VI) on the surfaces of the adsorbent, with fitting coefficients exceeding 0.99. Under the reaction conditions at pH = 2, the corresponding maximum adsorption capacities reached 111.23 mg/g, 125.00 mg/g, and 147.93 mg/g at different temperatures with 25 °C, 35 °C, and 45 °C, respectively. Langmuir model is well-suited for describing a uniform single adsorption process, where each molecule possesses constant thermal energy and an equal number of adsorption points [45]. The correlation coefficient of the Langmuir is highest at all three temperatures. This indicates that the adsorption process is mainly monolayer adsorption, and Cr (VI) is monolayer adsorption on the surface of the material. Secondly, the characteristic coefficient of the reaction adsorption degree (K_F_) is positively correlated with the adsorption amount. As the temperature increases, the K_F_ increases, indicating that the increase in the reaction temperature is conducive to the adsorption of Cr (VI) in the solution [46]. 

Temperature is a critical factor influencing the adsorption capacity of the solid–liquid phase medium [47]. In this study, the impact of temperatures of 298 K, 308 K, and 318 K on Cr (VI) with CTS@nZVI-WSPC was investigated. The findings indicated that the adsorption amount rises with increasing temperature, suggesting an endothermic nature of the adsorption process. The outcomes of the thermodynamic calculations are detailed in Appendix A.

Based on the data, the free energy decreases as the temperature rises, indicating higher adsorption efficiency at elevated temperatures. Both the enthalpy and entropy changes are positive, signifying an endothermic adsorption process that can spontaneously occur at higher temperatures. The increased degree of disorder in the reaction enhances structural disturbance during the adsorption process, facilitating solid–liquid phase contact and thereby augmenting the adsorption capacity.

### 3.4. Adsorption Kinetics

The investigation into adsorption kinetics provides insights into the rate of adsorption and the primary steps involved in the confinement of the adsorbent. As depicted in Appendix A, the adsorption rate of CTS@nZVI-WSPC was notably swift in the initial stages of the reaction. The adsorption kinetics at different temperatures showed the same trend. At 90 min, the adsorption rate did not change over time. At this time, the adsorption capacity at 25 °C, 35 °C, and 45 °C reached 41.06 mg/g, 43.23 mg/g, and 48.59 mg/g, respectively. This suggests that a substantial portion of adsorption occurs in the early phase of the adsorption process. Subsequently, owing to the reduction in available adsorption sites, the adsorption capacity of the adsorbent ceases to improve, ultimately reaching saturation.

In this experiment, several kinetic models were used to analyze the adsorption behavior of Cr (VI). As can be seen, the highest R^2^ was achieved by fitting data with the Pseudo-second model (Table 2), with the fitted Q_e_ value basically matching the actual value. Accordingly, we believed that the adsorption reaction process of Cr (VI) was mainly dominated by physical and chemical adsorption [48].

To elucidate the adsorption mechanism comprehensively, the intra-particulate diffusion model was employed to delineate the adsorption process in greater detail. The fitting results revealed the presence of boundary layer diffusion, intra-particle diffusion, and adsorption equilibrium throughout the adsorption of Cr (VI) by the CTS@nZVI-WSPC, signifying that internal diffusion does not singularly govern the adsorption process. A higher slope in the diffusion process within the particles indicates swifter boundary layer diffusion [49,50]. Notably, during the initial 60 min of the first stage, the slopes of the linear fitting curve were 1.95, 2.23, and 1.91, significantly larger than the subsequent two stages. This stage can be interpreted as the rapid diffusion stage, characterized by the driving force stemming from the substantial concentration difference between solid-liquid phase Cr (VI) and the porous structure of the composite material. This results in rapid adsorption under physical adsorption, driven by electrostatic action. The second stage (60–1440 min) was the slow adsorption stage, marked by a diminishing adsorption rate. Particle diffusion emerged as the predominant factor limiting the adsorption rate. As the adsorption reaction progressed, the adsorption sites approached saturation, leading to a continuous decline in the rate until equilibrium was reached [51].

### 3.5. Reusability of CTS@nZVI-WSPC

To investigate the reuse capacity of CTS@nZVI-WSPC, we collected the adsorbent particles using magnets at the end of each reaction, washed them with water, and dried them before subsequent use. As shown in Figure 5a, the removal rates for five consecutive runs were 95.8%, 90.36%, 85.11%, 78.37%, and 70.29%, respectively. These results demonstrate that CTS@nZVI-WSPC is effectively reusable. The removal efficiency in the first cycle is higher due to the availability of active sites. However, in subsequent runs, the number of binding sites decreases. Additionally, the removal efficiency declines in successive cycles due to the oxidation of nZVI and the formation of a passivation layer. It was noteworthy that the main form of Fe in the acidic system where the reaction proceeds was Fe (III) (Figure 5b), which may be due to the rapid oxidation of Fe (II) by reducing Cr (VI) [52,53]. Likewise, the leaching concentration of iron was relatively stable, and it may be related to the certain wrapping properties of chitosan.

### 3.6. XPS Analysis

XPS analysis results are presented in Figure 6a–d. Comparison of the total spectra before and after the adsorption reaction reveals distinct Cr characteristic peaks, predominantly located near 577 eV and 585 eV. This observation suggests that the majority of the Cr(VI) species involved in the reaction have undergone reduction to Cr(III) [54]. Figure 6b provided an in-depth analysis and comparative study of the Fe XPS spectra pre-reaction. The electron energy spectrum exhibits original Fe characteristic peaks primarily at approximately 711 eV and 724 eV [55]. Following deconvolution, four resolved peaks emerge at 710.28 eV, 723.7 eV, 712.06 eV, and 725.51 eV, corresponding to Fe2p3/2 and Fe2p1/2 orbitals for Fe(II), as well as Fe2p3/2 and Fe2p1/2 orbitals for Fe(III). The compositional analysis based on fitted peak areas reveals Fe(II) and Fe(III) contents of 50.95% and 49.05%, respectively. After the reaction, the Fe2p3/2 orbital of Fe(II) at 710.28 eV undergoes a leftward shift to 711.06 eV, indicative of Fe(II) consumption during the reduction reaction of Cr(VI). Consequently, the Fe(II) content decreases to 36.24% post-reaction compared to its initial level, thereby corroborating the aforementioned deductions.

As shown in Figure 6c, the characteristic peaks of O1s appear at the 530.7 eV and 532.2 eV binding energies, respectively, indicating that oxygen exists mainly in the form of O2-and [56]. After the adsorption reaction, the two oxygen peaks experience a shift of about 0.7 eV; this shift indicates that the oxygen-active group participates in the redox reaction between Fe and Cr. The characteristic main peaks of Cr that appeared after the reaction correspond to 577 ev and 585 ev, respectively [57]. After peak division processing, we can see that the characteristic peaks of Cr (III) 2p3/2 orbit appear near 576.73 ev and 577.72 ev, and the characteristic peaks of Cr (III) 2p1/2 orbit appear near 586.47 ev and 586.12 ev, respectively. Moreover, the peak strength signal of trivalent chromium is relatively significant, indicating that the reduction of hexavalent chromium has occurred significantly (Figure 6d).

### 3.7. Discussion of Adsorption Mechanism

Based on the relevant experimental results and fitting analysis, we assert that CTS@nZVI-WSPC represents a novel green biomass-modified adsorption material. The adsorption mechanism targeting Cr (VI) in water is postulated to encompass pore adsorption, chemical reduction, chelation reactions, and electrostatic interactions [58], as shown in Figure 7.

The adsorption efficiency of wheat straw biochar (WSPC) was relatively limited, primarily attributed to the high carbon content and pronounced aromaticity developed during the high-temperature carbonization process of wheat straw. Furthermore, the wheat straw employed in this experiment underwent carbonization with the introduction of KOH as a modifier, proving advantageous not only in enhancing the biochar’s pore structure but also in incorporating hydroxyl groups into the material’s framework, as substantiated by Fourier infrared photoskin analysis [59]. As shown in Figure 7, CTS@nZVI-WSPC has a developed pore structure and can provide sufficient adsorption points for Cr (VI).

During the adsorption process, nanoscale zero-valent iron (nZVI) plays a crucial role in reducing the toxicity of Cr (VI) owing to its robust reduction capacity. Concurrently, Fe (II) generated in the reduction process also contributes to the reduction of Cr (VI) [60]. The newly formed Cr (III), with an atomic radius nearly identical to Fe (II), tends to precipitate onto the nanomaterial’s surface in the form of complexes [61,62]. Additionally, a normalization reaction might occur between nZVI and Fe (III), generating more Fe (II), thereby enhancing the thoroughness of the reaction process and improving the material’s adsorption performance to some extent. Furthermore, chitosan served as a stabilizer in the preparation of nanoscale zero-valent iron (nZVI), significantly enhancing its dispersity and preventing aggregation. Through effective modification, there was a significant increase in the adsorption capacity with the new adsorbent CTS@nZVI-WSPC, and the composite exhibited optimal adsorption performance at lower pH values. The presence of more coordination heteroatoms (-OH, -NH) provided by chitosan further improved the adsorption performance compared to the other two materials.

## 4. Conclusions

Utilizing wheat straw as the raw material, an amino-functionalized material was synthesized through a stepwise modification process, presenting itself as a potential alternative adsorbent for Cr (VI) in water. In comparison to a singular biochar, the modified material exhibits superior reduction and adsorption performance. Based on the above research, we can infer that the composite material of CTS@nZVI-WSPC may become a reliable adsorbent for environmental remediation due to its green, low-cost, and excellent adsorption performance. However, they still face many challenges in large-scale applications. For example, the synthesis of biochar still relies on high-temperature calcination methods, which have high energy consumption and are not conducive to large-scale use. Additionally, in the discussion of mechanism synthesis and limiting steps, more in-depth discussions are needed, which will be our focus in future efforts.

## Figures and Tables

**Figure 1 nanomaterials-14-00973-f001:**
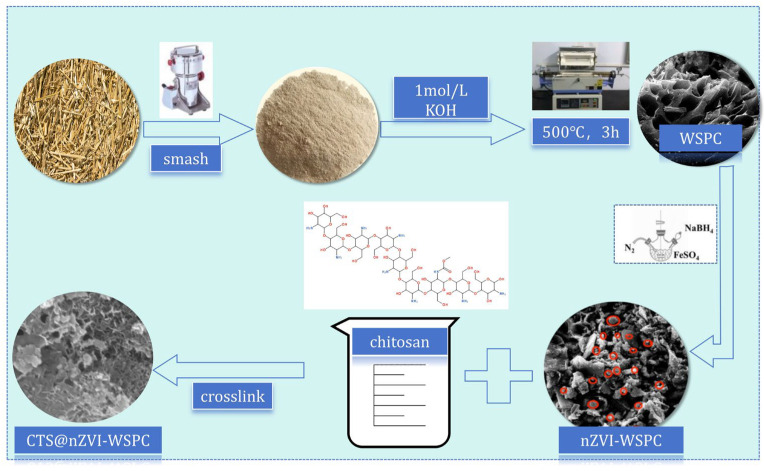
Schematic diagram of the preparation process of three adsorbent materials.

**Figure 2 nanomaterials-14-00973-f002:**
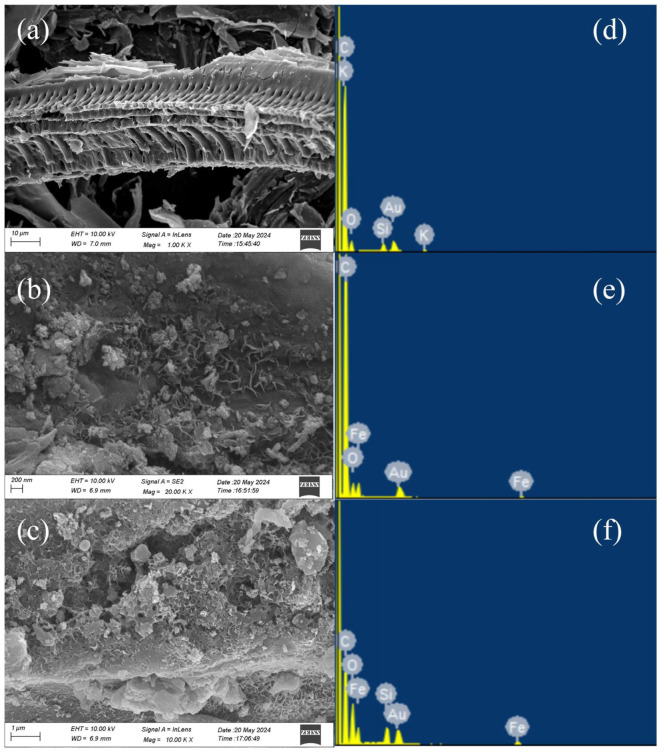
SEM and EDS image of three materials (**a**) SEM of WSPC; (**b**) SEM of nZVI-WSPC; (**c**) SEM of CTS@nZVI-WSPC; (**d**) EDS of WSPC; (**e**) EDS of nZVI-WSPC; (**f**) EDS of CTS@nZVI-WSPC.

**Figure 3 nanomaterials-14-00973-f003:**
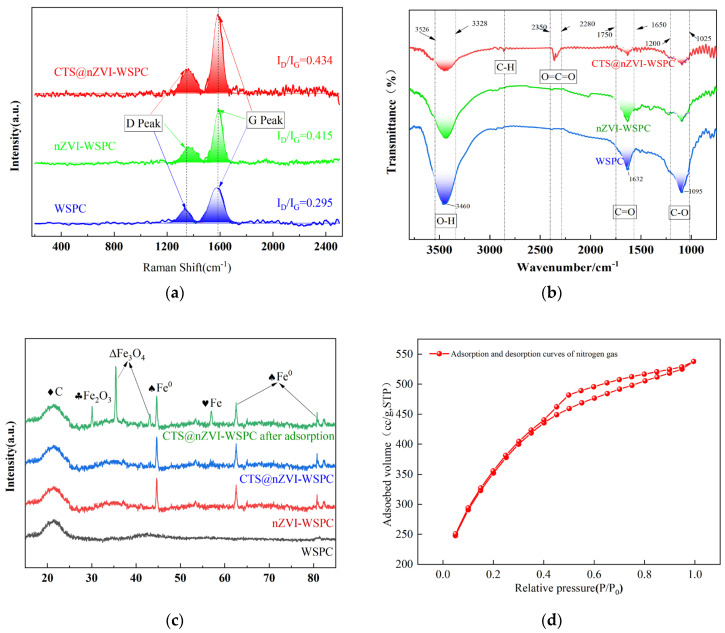
Characterization results of three materials: (**a**) The Raman analysis map; (**b**) FTIR curves; (**c**) XRD patterns; (**d**) N_2_ adsorption-desorption isotherms of CTS@nZVI-WSPC.

**Figure 4 nanomaterials-14-00973-f004:**
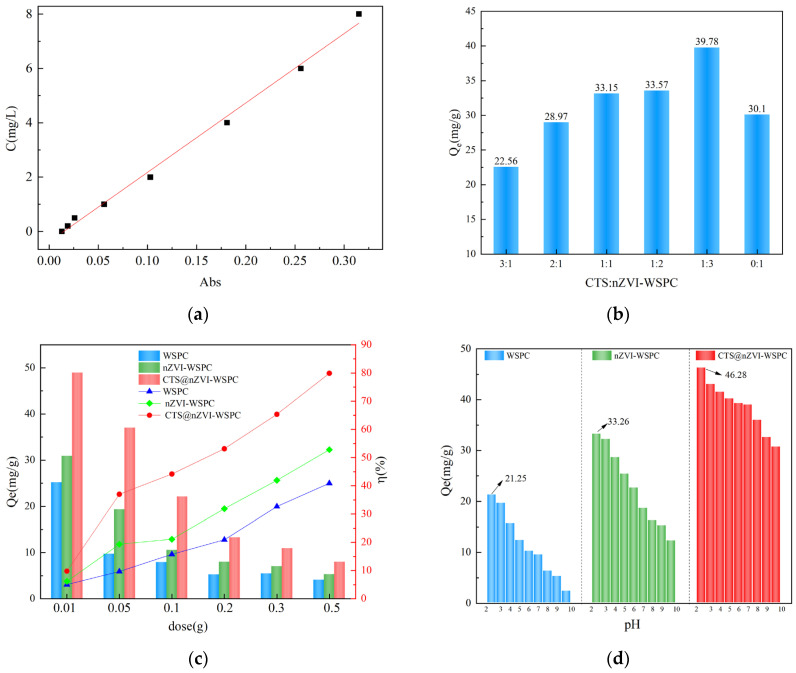
(**a**) Standard curve for hexavalent chromium determination; (**b**) ratio of CTS to nZVI-WSPC (C_0_ = 50 mg/L, T = 25 °C, t = 24 h, dose = 0.05 g, pH = 7.0); (**c**) dose of adsorbent (C_0_ = 50 mg/L, T = 25 °C, t = 24 h, pH = 7.0); (**d**) pH (C_0_ = 50 mg/L, T = 25 °C, t = 24 h, dose = 0.05 g).

**Figure 5 nanomaterials-14-00973-f005:**
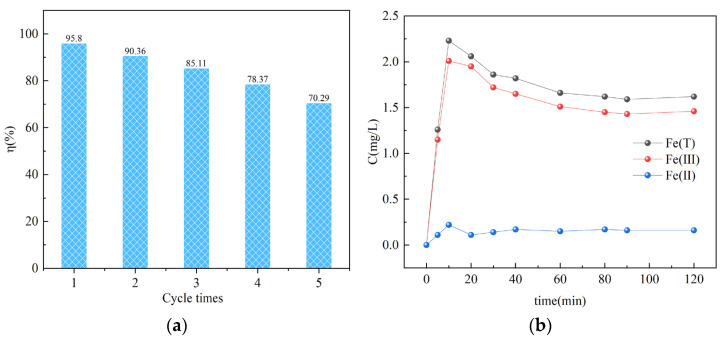
(**a**) Removal efficiency of CTS@nZVI-WSPC composite after five uses; (**b**) Changes in Fe concentrations during the reaction (condition: C_0_ = 50 mg/L, T = 25, t = 24 h, dose = 0.05 g, pH = 2).

**Figure 6 nanomaterials-14-00973-f006:**
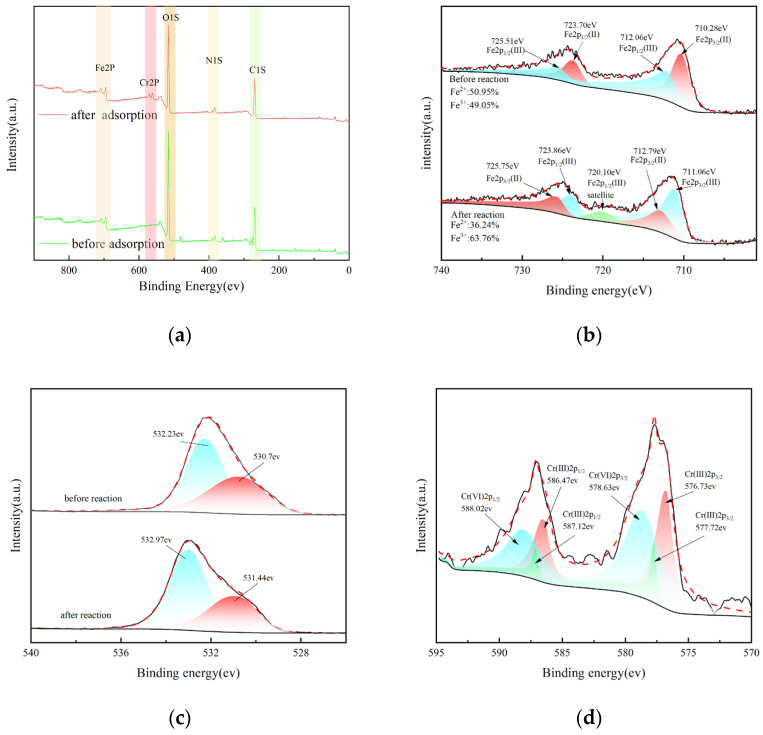
(**a**) Wide scan XPS analysis before and after reaction; (**b**) narrow scan of Fe before and after reaction; (**c**) narrow scan of O before and after reaction; (**d**) narrow scan of Cr after reaction.

**Figure 7 nanomaterials-14-00973-f007:**
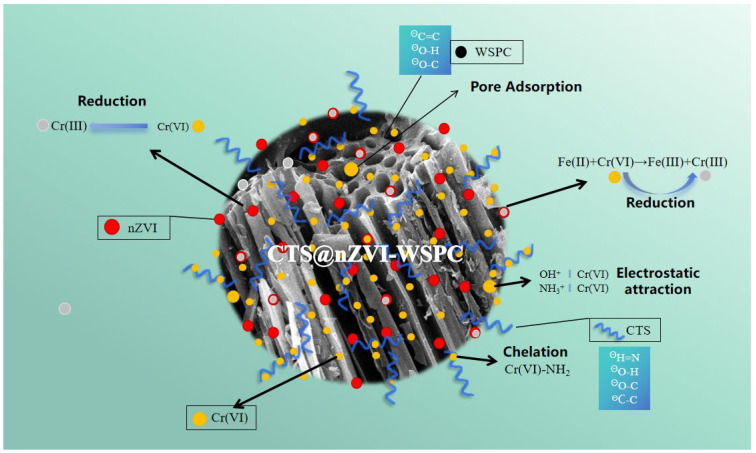
Diagram of the adsorption reaction mechanism.

**Table 1 nanomaterials-14-00973-t001:** Adsorption isotherm fitting parameters of different materials.

		Langmuir			Freundlich			Temkin		
		Q_m_ mg/g	*K_L_* L/mg	R^2^	*K_F_*	*n*	R^2^	*A*	*B*	R^2^
25 °C		111.23	0.0037	0.9920	12.116	2.1683	0.9769	1.3573	17.1767	0.9074
35 °C	CTS@nZVI-WSPC	125.00	0.0405	0.9956	12.965	2.2762	0.9795	1.3671	19.9112	0.9242
45 °C		147.93	0.0686	0.9821	14.532	1.8126	0.9580	1.3788	25.1972	0.9572

**Table 2 nanomaterials-14-00973-t002:** Adsorption kinetics fitting parameters of CTS@nZVI-WSPC.

		Pseudo-First-Order			Pseudo-Second-Order			Intra-Particle	
		Q_e_ mg/g	*K*_1_ L/mg	R^2^	Q_e_ mg/g	*K_2_*	R^2^	*Kp* _1_	*Kp* _2_
25 °C		42.826	0.1410	0.9611	41.271	0.0047	0.9999	1.95	0.002
35 °C	CTS@nZVI-WSPC	44.511	0.1468	0.9309	44.762	0.0054	0.9999	2.23	0.015
45 °C		47.816	0.2088	0.9649	47.984	0.0089	0.9999	1.91	0.049

## Data Availability

Data are contained within the article and Appendix A.

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
