# Peer review of "Enhanced Adsorptivity of Hexavalent Chromium in Aqueous Solutions Using CTS@nZVI Modified Wheat Straw-Derived Porous Carbon"

_nanomaterials, 2024, doi:10.3390/nano14110973_

Round 1

Reviewer 1 Report

Comments and Suggestions for Authors

1. The SEM of nZVI-WSPC results in Fig.2B show nZVI with micron size instead of nano, and discovering a significant presence of iron in oxide form in Fig.3C and Fig.7D. Thus, the naming of nZVI is unreasonable, or please provide evidence of nano.

2.Considering the prevalence of iron oxides, it is crucial to evaluate stability of composite and the potential for iron leaching, especially under acidic conditions. Hence, it's imperative to ascertain the concentration of iron ions in the solution after adsorption. Moreover, the XRD analysis of composite after adsorption should be measured to verify stability.

3. Cr concentration were tested by UV spectrophotometry, whereas the standard curve has not been provided. 

4. Fig.3A X axis, Raman not Roman. Figure 3c, font too small for axis labels and legend. Make all font consistent size for all figure panels. There is room in the figure to make the figures bigger so there is less space in between the figures. 

5. The removal rate of three adsorbent materials in Fig.4B suggest that the adsorption efficiency is not good enough.

6. The adsorption hysteresis phenomenon was observed by BET measurement in Fig.7A, which indicated the Type IV isotherm with an H4 hysteresis loop.

7. The Adsorption model (Langmuir) does not match the mechanism in Fig.8. 

8. Does the CTS@nZVI-WSPC's adsorption process, which involves chemical transformation (described in manuscript line 521-523), imply its disposable?

9. For the Abstract, the CTS@nZVI-WSPC was introduced before the CTS, so it is confusing to know what the abbreviation stands for before reading the CTS abbreviation.

10. Chemical reagents: Check capitalization of chemical names. Capitalization not necessary. The names of chemicals are not capitalized unless they are the first word of a sentence

11. In the text authors describe FTIR peak 2280-2352 cm-1 as NH2, but on the Figure it is labeled as CO2.

12. Line 360: NH3+ should be NH3+

13. Put a space after a period (.)

14 .May want to increase the size of the panels in figure 4 to make better use of space in the figure.

15. Section 3.5: formatting the text for spaces and putting a “-“ between word breaks that are broken between lines.

16. Figure 5: make panels bigger to better use the space.

Comments on the Quality of English Language

editing for minor issues such as spacing, capitals should be performed

Author Response

Thank you very much to the very professional, serious and responsible reviewers for their valuable comments on this paper. After this period of experimental supplement and further improvement of the paper, we have finally completed the relevant revision. Now the suggestions are explained one by one.

The review comments 1 and the revision instructions are as follows:

1.The SEM of nZVI-WSPC results in Fig.2B show nZVI with micron size instead of nano, and discovering a significant presence of iron in oxide form in Fig.3C and Fig.7D. Thus, the naming of nZVI is unreasonable, or please provide evidence of nano.

Re: We reapplied the new SEM for higher resolution identification and observation. We can see that Fe mainly exists on the surface of the composite material in the form of spherical or irregular spherical particles, and the new SEM can prove the presence of nanoscale iron, see Figure 2 at line 192. In fact, the Fe size is present at both the nanometer and micrometer scale due to the heterogeneity of the material during synthesis. In addition, the reference liquid phase reduction method for the synthesis of materials is a proven effective and mature synthesis method of nano-valent iron, which is called nano-valent iron in the literature. Some references are added here, see reference [24][39][49] et al. XRD shows that the presence of Fe is in a variety of crystal structures, mainly because nano-zero-valent iron materials are prone to oxidation to form iron oxides. However, the diffraction peak of XRD near 44.5 is sufficient to prove the existence of zero-valent iron, which has been mentioned in many literature. Regarding this issue, we characterized XRD in a short time of resomaterial synthesis, which confirmed the existence of zero-valent iron. The oxide form in XPS is present because the zero-valent iron participates in the redox reaction.

2.Considering the prevalence of iron oxides, it is crucial to evaluate stability of composite and the potential for iron leaching, especially under acidic conditions. Hence, it's imperative to ascertain the concentration of iron ions in the solution after adsorption. Moreover, the XRD analysis of composite after adsorption should be measured to verify stability.

Reply: We supplemented the repeatability test of composite materials and the leaching experiment of Fe under acidic conditions, and the results proved that the removal rate ofCr with 50 mg/L could still reach more than 70% after five consecutive cycles. Besides, the concentration of Fe was stable at 1.5 mg/L after 120min under acidic conditions, and the leaching rate was not high. This is because we wrapped the nano-zero-valent iron with chitosan to make it more stable. In addition, we conducted a new XRD determination of the material, and supplemented the XRD analysis of the composite material after adsorption. It can be seen that the presence of Fe before the reaction is more abundant, which also verifies the oxidation reaction of zero-valent iron in the reaction system, generating Fe2O3 and Fe3O4, as can be seen from Figure 3 (c).

3.Cr concentration were tested by UV spectrophotometry, whereas the standard curve has not been provided.

Reply: The standard curve for Cr is provided in the Supplementary Material. Detailed results can be seen in Figure4(a).

  1. Figure 3c, font too small for axis labels and legend. Make all font consistent size for all figure panels. There is room in the figure to make the figures bigger so there is less space in between the figures. 

Reply: For this common problem, the full drawings have been redrawn to make them look clearer.

5.The removal rate of three adsorbent materials in Fig.4B suggest that the adsorption efficiency is not good enough.

Reply: In Figure 4b, the removal rate of Cr (VI) of the three materials in the solution is compared to show that the adsorption rate of the composite materials modified by nano-zero-valent iron and chitosan is significantly higher than that of a single biochar material. It is worth mentioning that we did not adjust the pH of the reaction system at this time, so the removal rate has not reached the best. The issue has been supplementary explained in the paper.

6.Fig.7A, which indicated the Type IV isotherm with an H4 hysteresis loop.

Reply: The adsorption hysteresis was observed by BET measurements in Figure 7a, indicating that the type IV isotherm is a type h4 isotherm. To illustrate the properties of the material, we put the results of BET into the characterization section. According to the test results, the curve is more consistent with the type IV isotherm, which has been further explained in the text. Detailed description can be seen on lines 252 to 259.

  1. The Adsorption model (Langmuir) does not match the mechanism in Fig.8. 

Reply: We have refined and elaborated the mechanism of the whole reaction through further combing. The Langmuir model shows that the removal process is closer to the monolayer ion coverage, mainly at sites with the same energy on the adsorbent surface. Through kinetic analysis, the whole adsorption process can be obtained more in line with the quasi-secondary model, which indicates that chemical adsorption occupies a certain role in the reaction process. In addition, the results of XPS can also verify this conclusion. We combine the paper data and the characterization results to reorganize the process and update the reaction mechanism diagram.

  1. Does the CTS@nZVI-WSPC's adsorption process, which involves chemical transformation (described in manuscript line 521-523), imply its disposable?

Re: In order to verify the reuse performance of the material, we supplemented the relevant experiments, and the results proved that the removal efficiency of the material was reduced after 5 uses, but not one-time. The detailed analysis results can be seen from Figure 5 (a), although there is chemical adsorption in the process, there are also reaction mechanisms such as pore adsorption and electrostatic adsorption.

  1. For the Abstract, the CTS@nZVI-WSPC was introduced before the CTS, so it is confusing to know what the abbreviation stands for before reading the CTS abbreviation.

Re: For this question, we will provide supplementary instructions where the first CTS appeared, which can be seen in line 17 of the latest abstract.

  1. Chemical reagents: Check capitalization of chemical names. Capitalization not necessary. The names of chemicals are not capitalized unless they are the first word of a sentence

Reply: The question has been corrected in the text. Details can be seen in the Supplementary material.

11.In the text authors describe FTIR peak 2280-2352 cm-1 as NH2, but on the Figure it is labeled as CO2.

Reply: This is a writing error. After checking and literature review and comparison, the FTIR peak 2280-2352 cm-1 is CO2, which has been revised in the text. Can be seen in the analysis of the relevant FTIR.

  1. Line 360: NH3+ should be NH3+

Re: The error has been revised and can be seen in detail at line 319.

  1. Put a space after a period (.)

Reply: The error has been revised.

  1. May want to increase the size of the panels in figure 4 to make better use of space in the figure.

Reply: The drawings in the full text have been adjusted.

  1. Section 3.5: formatting the text for spaces and putting a “-“ between word breaks that are broken between lines.

Reply: Similar questions have been adjusted throughout the article.

16. Figure 5: make panels bigger to better use the space.

Reply: Similar questions have been adjusted throughout the article.

Reviewer 2 Report

Comments and Suggestions for Authors

Enhanced adsorptivity of hexavalent chromium in aqueous solutions using CTS@nZVI modified wheat straw-derived porous carbon

Manuscript number: nanomaterials-3013617

The work has significant drawbacks and it is necessary for the authors to improve manuscript with addition of new data in order to be reconsidered for publication in Nanomaterilas.

1. Well known equations 1-10 should be transferred to supplementary material.

2. In Figure 1 the structure of chitosan is unclear (higher resolution is necessary).

3. Check statement “2g of wheat straw biochar (WSPC) was dissolved in 50mL of FeSO4·7H2O solution”.

4. Authors should clarify sections: “2.2.1 Single-factor batch experiment” and “2.2.2 Absorption Isotherm experiment”. What authors mean for “Reaction conditions” in Table 1.

5. “This section may be divided by subheadings. It should provide a concise and precise description of the experimental results, their interpretation, as well as the experimental conclusions that can be drawn.”? Whoever wrote this it is good suggestions what authors should do. The manuscript should be corrected and edited in detail.

6. Resolution of Figures 3, 5, 6 and 7 is low. Please enhance the clarity of the graphs, and ensure they are well-labeled and easy to understand.

7. Explain presence of phase “C60”. Also, authors talked about adsorptivity of nZVI but no discussion on Fe2O3 phases contribution was included.

8. Please clarify what is optimal operational condition: “Under the reaction conditions at pH=2, the corresponding maximum adsorption capacities reached 111.23 mg/g, 125.00mg/g, and 147.93 mg/g at different temperature with 2535and 45, respectively.” If it is pH 2, author should analyze adsorbent stability, leaching of ions etc. Also, it is not appropriate to use term “reaction” (example Figure 7) but adsorption where oxidation/reduction processes take place in the course of adsorption. These processes need deeper analysis (page 17, lines 498-506).

If the author claim Alternatively, Cr (VI) may bind to oxygen-containing functional groups, forming Cr(OH)3 precipitation [58][59]” it is necessary to prove structure of adsorbent surface. This is closely related to desorption and adsorption in new cycle (results not presented).

Also, more effective discussion could be based on adsorption results and mechanisms related to constituent of CTS: nZVI-WSPC.

9. Statistical analysis of experimental data (considering tabular and graphical presentation of experimental data)?

10. Isoelectric point or pHpzc is not presented?

11. English improvement is necessary. Some sections are overly technical and could be simplified for broader accessibility. Aim for clear, concise language.

12.  Please strengthen the literature review and expand on how this work builds upon or differs from existing research, emphasizing the unique contribution. A more comprehensive discussion on the limitations of the current study and potential areas for future research would strengthen the manuscript. Consider concept sustainability.

Comments on the Quality of English Language

English improvement is necessary. Some sections are overly technical and could be simplified for broader accessibility. Aim for clear, concise language.

Author Response

Thank you very much to the very professional, serious and responsible reviewers for their valuable comments on this paper. After this period of experimental supplement and further improvement of the paper, we have finally completed the relevant revision. Now the suggestions are explained one by one.

1.Well known equations 1-10 should be transferred to supplementary material.

Reply: The supplementary materials of this paper have been combed, and the relevant formulas and pictures have been explained

2.In Figure 1 the structure of chitosan is unclear (higher resolution is necessary).

Re: We conducted a characterization analysis of the material and redrew Figure 1 to make the picture clearer.

3.Check statement “2g of wheat straw biochar (WSPC) was dissolved in 50mL of FeSO4·7H2O solution”.

Re: The statement here is not accurate, and the biochar is insoluble. Therefore, the description was changed to 2g of wheat straw biochar (WSPC) dispersed in 50 mL of FeSO4·7H2O solution.It can be seen at line 115 of the text.

4.Authors should clarify sections: “2.2.1 Single-factor batch experiment” and “2.2.2 Absorption Isotherm experiment”. What authors mean for “Reaction conditions” in Table 1.

Reply: The original statement about the reaction conditions in Table 1 is ambiguous. For this reason, we adjusted and sorted out the overall framework of the article based on the next suggestion, and refined the conditions of the experiment in each section.

5.“This section may be divided by subheadings. It should provide a concise and precise description of the experimental results, their interpretation, as well as the experimental conclusions that can be drawn.”? Whoever wrote this it is good suggestions what authors should do. The manuscript should be corrected and edited in detail.

Reply: For this question, we reorganized the framework of the paper, and described it according to the characterization results, single-factor batch experiments and isothermal adsorption experiments, adsorption kinetic experiments, etc. Make the whole regulation look clearer.

6.Resolution of Figures 3, 5, 6 and 7 is low. Please enhance the clarity of the graphs, and ensure they are well-labeled and easy to understand.

Reply: the full text of the picture has been replaced to make the picture more clear.

7.Explain presence of phase “C60”. Also, authors talked about adsorptivity of nZVI but no discussion on Fe2O3phases contribution was included.

Reply: XRD shows an obvious signal near the diffraction peak of 25°, which can be determined as the characteristic peak of carbon through relevant literature review and PDF standard card comparison, which also verifies the rationality of the existence of biochar. The nZVI, which is loaded on the biochar surface as a modified material, participates in the redox reaction of Cr and forms Fe2O3 in the process, which is illustrated in detail in the analytical section of XPS.

8.Please clarify what is optimal operational condition: “Under the reaction conditions at pH=2, the corresponding maximum adsorption capacities reached 111.23 mg/g, 125.00mg/g, and 147.93 mg/g at different temperature with 25℃、35℃and 45℃, respectively.” If it is pH 2, author should analyze adsorbent stability, leaching of ions etc. Also, it is not appropriate to use term “reaction” (example Figure 7) but adsorption where oxidation/reduction processes take place in the course of adsorption. These processes need deeper analysis (page 17, lines 498-506).If the author claim “Alternatively, Cr (VI) may bind to oxygen-containing functional groups, forming Cr(OH)3 precipitation [58][59]” it is necessary to prove structure of adsorbent surface. This is closely related to desorption and adsorption in new cycle (results not presented).Also, more effective discussion could be based on adsorption results and mechanisms related to constituent of CTS: nZVI-WSPC.

Reply: The best operating conditions have been added, such as the amount of adsorbent and reaction temperature. In addition, the results of the material, the results of the entry and exit of Fe under acidic conditions, proved the stability of the material. Detailed instructions can be seen in the Figure5. In the reaction mechanism, through literature reading and combined with the experimental results of this paper, four mechanisms of pore adsorption,oxidation/reduction, electrostatic adsorption and chelation reaction. This is explained in detail in the Discussion section.

9.Statistical analysis of experimental data (considering tabular and graphical presentation of experimental data)?

Reply: The experimental data in this paper are the mean of three repeated tests, and the effective digit expression of some data is not accurate, which has been revised in the paper.

10.Isoelectric point or pHpzc is not presented?

Reply: The determination of potential is helpful to analyze whether the experimental process involves the mechanism of electrostatic adsorption. The research on this aspect is more common and the consensus is formed, so the determination of potential is not conducted in this paper. We compare and cite the literature conclusions of similar studies to illustrate this problem. Details can be seen in lines 299 to 307 in the text.

11.English improvement is necessary. Some sections are overly technical and could be simplified for broader accessibility. Aim for clear, concise language.

Reply: We have polished and improved the English expression of the whole paper to make it more clear and concise.

12.Please strengthen the literature review and expand on how this work builds upon or differs from existing research, emphasizing the unique contribution. A more comprehensive discussion on the limitations of the current study and potential areas for future research would strengthen the manuscript. Consider concept sustainability.

Reply: In the literature review section, we further combed through the removal application of Cr in the early stage, emphasizing the uniqueness and contribution of this study. And complements the potential future research directions to make the review section more comprehensive and in-depth.

Reviewer 3 Report

Comments and Suggestions for Authors

Comments from Reviewer

Manuscript ID: nanomaterials-3013617-peer-review-v1

Title: Enhanced adsorptivity of hexavalent chromium in aqueous solutions using CTS@nZVI modified wheat straw -derived porous carbon

The current form's presentation of methods and scientific results is satisfactory for publication in the Nanomaterials journal. Some comments apply to the entire article. Please take this into account when making corrections. Compared to the previous version, you can see a considerable improvement. The minor and significant drawbacks to be addressed can be specified as follows:
1.    Line 18, “BET” is not the technique – BET is the theoretical model or the theoretical equations. Writing “N2 adsorption (T=77K)” is better. See also line 97.
2.    I suggest writing an abstract with descriptive and concise information about this research. A standard abstract definition is: “An abstract is a concise summary of an experiment or research project. It should be brief -- typically under 200 words. The purpose of the abstract is to summarize the research paper by stating the purpose of the research, the experimental method, the findings, and the conclusions.”
3.    Lines 52 and 53. Problems with “,”, i.e., comma.
4.    Line 115, “Those three absorbents were obtained as shown in Figure 1.” What adsorbents?
5.    Line 166. Isotherm ---> isotherm.
6.     “absorption” or “adsorption”. See lines 166 and 370. Absorption ---> adsorption. Check all the manuscript.
7.    2.2.2 Absorption Isotherm experiment. Equipment? See also 2.2.3 and 2.2.4
8.    2.3 Characterization. Equipment for the low-temperature nitrogen adsorption/desorption data.
9.    Fig. 2. Double numbering of panels. What for?
10.    Fig. 3(c). The legend is unreadable - the font is too small
11.    Lines 260 and 407, “correlation coefficients (R2)” R – the correlation coefficient and R2 – the termination coefficient.
12.    Some abbreviations are explained several times in the text. It is sufficient to do it the first time. See, for example, R2.
13.    Line 447. The explanation of BET is too late. Please explain the abbreviation the first time you use it.
14.    Fig. 7(a). This is not an isotherm of the first type, as indicated by the shape of the isotherm and hysteresis loop.
15.    Fig. 7(b). What method was chosen for the pore distribution? BJH? What is the lower range of analyzed pores?

Sincerely,
    The reviewer.

Author Response

Thank you very much to the very professional, serious and responsible reviewers for their valuable comments on this paper. After this period of experimental supplement and further improvement of the paper, we have finally completed the relevant revision. Now the suggestions are explained one by one.

1.Line 18, “BET” is not the technique – BET is the theoretical model or the theoretical equations. Writing “N2 adsorption (T=77K)” is better. See also line 97.

Reply: The BET presentation has been modified.

2.I suggest writing an abstract with descriptive and concise information about this research. A standard abstract definition is: “An abstract is a concise summary of an experiment or research project. It should be brief -- typically under 200 words. The purpose of the abstract is to summarize the research paper by stating the purpose of the research, the experimental method, the findings, and the conclusions.”

Reply: After discussion, we wrote the abstract again, focusing on the methods and conclusions of the research, and making it more concise and clear.

  1. Lines 52 and 53. Problems with “,”, i.e., comma.

Reply: It was revised.

  1. Line 115, “Those three absorbents were obtained as shown in Figure 1.” What adsorbents?

Reply: These three adsorbents refer to WSPC, nZVI-WSPC, and CTS @ nZVI-WSPC, which have adjusted the position of the sentence and described the three materials, which can be seen in detail in line 133.

  1. Line 166. Isotherm ---> isotherm.

Reply: It was revised. Details can be seen at line 153

  1. “absorption” or “adsorption”. See lines 166 and 370. Absorption ---> adsorption. Check all the manuscript.

Reply: The full text has been revised to address this issue.

7.2.2.2 Absorption Isotherm experiment. Equipment? See also 2.2.3 and 2.2.4

Reply: I have adjusted the overall framework of the paper to make it clearer. The relevant instruments and equipment used are described in the Supplementary Material.

8.2.3 Characterization. Equipment for the low-temperature nitrogen adsorption/desorption data.g of panels. What for?

Re: The equipment used for nitrogen adsorption and desorption has been described in the supplementary material.

9.Fig. 2. Double numbering of panels. What for?

Re: The part displayed repeatedly has been changed.

10.Fig. 3(c). The legend is unreadable - the font is too small

Reply: The drawings in the full text have been revised.

11.Lines 260 and 407, “correlation coefficients (R2)” R – the correlation coefficient and R2 – the termination coefficient.

Reply: Revised, see line 204.

12.Some abbreviations are explained several times in the text. It is sufficient to do it the first time. See, for example, R2.

Reply: The full text has been checked and the repeated explanations have been deleted.

13.Line 447. The explanation of BET is too late. Please explain the abbreviation the first time you use it.

Reply: The BET results have been advanced to the characterization section and can be seen in Figure 3 (d).The BET results have been advanced to the characterization section and can be seen in Figure 3 (d).

14.7(a). This is not an isotherm of the first type, as indicated by the shape of the isotherm and hysteresis loop.

Reply: The adsorption hysteresis was observed by BET measurements in Figure 7a, indicating that the type IV isotherm is the type h4 isotherm. It has been revised. It can be seen at Article 252 in the text.

15.7(b). What method was chosen for the pore distribution? BJH? What is the lower range of analyzed pores?

Reply: We used the BJH method. This issue has been addressed and supplemented in the supplementary mater

Round 2

Reviewer 1 Report

Comments and Suggestions for Authors

Authors have corrected the manuscript.  However, there are many english corrections needed to the new text.

Line 16: "absorent" should be adsorbent, "Using" should be used.

Line 136: "adsoprtion" should be adsorption.

142: "Chosen of absorbent" should be Choice of adsorbent.

148: "Absorbent" should be adsorbent

182/444: "Dissuasion" should be discussion

277: "absorbent" should be adsorbent

please do thorough spell check as the above list might not be comprehensive.

Comments on the Quality of English Language

See above comments to author

Author Response

I have carefully reviewed and revised the entire manuscript based on the feedback from the reviewers, and the specific changes can be found in the revised version. 

Reviewer 2 Report

Comments and Suggestions for Authors

According to performed corrections the revised manuscript can be accepted

Comments on the Quality of English Language

minor correction is necessary

Author Response

(The authors gave the same response as above.)

Reviewer 3 Report

Comments and Suggestions for Authors

In my opinion the corrected work can be accepted for publication.

Author Response

I have carefully reviewed and revised the entire manuscript based on the feedback from the other reviewers, and the specific changes can be found in the revised version. Thank you very much for your suggestions and affirmation.